# Physicochemical Foundations of Life that Direct Evolution: Chance and Natural Selection are not Evolutionary Driving Forces

**DOI:** 10.3390/life10020007

**Published:** 2020-01-21

**Authors:** Didier Auboeuf

**Affiliations:** Laboratory of Biology and Modelling of the Cell, Univ Lyon, ENS de Lyon, Univ Claude Bernard, CNRS UMR 5239, INSERM U1210, 46 Allée d’Italie, Site Jacques Monod, F-69007 Lyon, France; didier.auboeuf@inserm.fr

**Keywords:** Evolution, Darwinism, biophysics, RNA, Origin of life

## Abstract

The current framework of evolutionary theory postulates that evolution relies on random mutations generating a diversity of phenotypes on which natural selection acts. This framework was established using a top-down approach as it originated from Darwinism, which is based on observations made of complex multicellular organisms and, then, modified to fit a DNA-centric view. In this article, it is argued that based on a bottom-up approach starting from the physicochemical properties of nucleic and amino acid polymers, we should reject the facts that (i) natural selection plays a dominant role in evolution and (ii) the probability of mutations is independent of the generated phenotype. It is shown that the adaptation of a phenotype to an environment does not correspond to organism fitness, but rather corresponds to maintaining the genome stability and integrity. In a stable environment, the phenotype maintains the stability of its originating genome and both (genome and phenotype) are reproduced identically. In an unstable environment (i.e., corresponding to variations in physicochemical parameters above a physiological range), the phenotype no longer maintains the stability of its originating genome, but instead influences its variations. Indeed, environment- and cellular-dependent physicochemical parameters define the probability of mutations in terms of frequency, nature, and location in a genome. Evolution is non-deterministic because it relies on probabilistic physicochemical rules, and evolution is driven by a bidirectional interplay between genome and phenotype in which the phenotype ensures the stability of its originating genome in a cellular and environmental physicochemical parameter-depending manner.

## 1. Introduction

The current framework of evolutionary theory that comes from the modern synthetic theory of evolution postulates that evolution relies on random mutations, generating a diversity of phenotypes on which natural selection acts. This concept is widely accepted in the scientific community despite the fact that some important issues concerning it have been raised. [1,2,3]. The notion of random mutations can lead to multiple interpretations. It can mean that the nature or the location of mutations is random, and indeed, mutations are often described as “errors” during replication [4,5]. However, many factors influence the rate, nature, and location of mutations [5,6,7,8,9,10]. Thus, the appearance of a mutation is probabilistic and depends on multiple cellular- and environment-dependent physicochemical parameters. The notion of random mutations can also mean that the probability of a mutation is independent of the phenotype it generates. One of the objectives of this article is to show that if cellular- and environment-dependent physicochemical parameters influence the frequency, nature, and location of mutations, the probability of a mutation should depend on the phenotype it generates. Indeed, a continuum between physiological and genetic adaptation is shown, as already has been proposed in [11,12,13], which indicates that physiological adaptation facilitates and guides genetic variations in an environment-depending manner (see below).

The notion of natural selection is also subject to multiple interpretations because it can be negative, positive, or neutral [14,15,16,17]. How can evolution be explained, if chance generates a large number of possibilities, and natural selection can be positive, negative, or neutral? In addition, the notion of natural selection is of limited interest, because a living organism observable over more or less long time periods is necessarily adapted to its environment; otherwise, it disappears without leaving any descendants. The second objective of this article is to replace the notion of natural selection with the notion that the role of the phenotype in evolution is to maintain the physicochemical integrity and stability of its originating genome. This establishes feedforward and feedback loops between the genome and the phenotype, i.e., the genome generates a phenotype that exerts a feedback loop by either maintaining genome stability or guiding genome variations depending on its efficiency to relax environment-dependent physicochemical constraints.

Another issue raised by the current model of evolution relies on the fact that the combination of chance and natural selection does not provide a single conceptual framework that simultaneously explains both evolution and organism activities, since evolution, but not organism activities, would rely on chance and natural selection. However, life and evolution are inseparable and must depend on the same fundamental principle. Indeed, living organisms are hierarchically structured since multicellular organisms are composed of cells that are composed of molecules. The lower levels of organization have necessarily appeared before more complex living forms and fundamental principles of evolution must be applicable from molecules to complex organisms. In this setting, the current model of evolution was established using a “top-down” approach originating from Darwinism that is based on observations made of complex multicellular organisms [18] which was modified after the discovery of DNA. In other words, there was no other choice than to define “chance” and “natural selection” as evolutionary driving forces in order to explain the emergence of complex phenotypes of multicellular organisms prior to understanding their molecular origin or the underlying physicochemical mechanisms. However, to uncover evolutionary driving forces, one needs to define them from the physicochemical processes at the molecular origin of life. Therefore, in this article, a “bottom-up” approach is used according to the facts that (i) life started with the emergence of nucleic and amino acid polymers before the emergence of more complex forms of life and (ii) physicochemical laws are the foundations of cellular processes and complex organism activities. A “bottom-up” approach redefines (i) the notion of chance in a precise context of physicochemical laws and (ii) the notion of adapted phenotype not in terms of organism fitness but rather in terms of the impact on genome stability.

## 2. Overview 

Life is based on two types of polymers, nucleic acid polymers (DNA and RNAs) and amino acid polymers (proteins). Over the last decades, the emphasis has been on a functional dichotomy in which nucleic acids are described as supporting genetic information, while proteins perform the cellular activities. As a consequence, nucleic acids are often “simply” considered as the carrier (DNA) or vectors (RNAs) of genetic information and are represented in the form of a suite of letters A, C, G, and T, corresponding to the four nucleotides that composed them. Thus, the physicochemical properties of nucleic acids are obscured in the context of the current synthetic theory of evolution, in which evolution corresponds basically to the random substitutions of one letter by another one. In Section 3, it is underlined that the physicochemical properties of nucleic and amino acid polymers depend on their composition, which is constrained by cellular- and environment-dependent physicochemical parameters. Next, it is shown that this principle has consequences for the way evolution proceeds.

Indeed, in Section 4, it is highlighted that the emergence of life probably corresponds to the establishment of the interdependency between RNAs (or similar molecules) and proteins (or similar molecules). This means that, as observed in modern organisms, protein synthesis depends on RNAs as templates and RNA synthesis depends on proteins. This interdependency generates a positive feedback loop, as an RNA (the proto-genome) generates a protein (the proto-phenotype) that contributes to the synthesis of the RNA on which it depends (Figure 1A). Moreover, RNAs are unstable and are rapidly degraded by hydrolysis. Therefore, the RNA/protein system can only self-reproduce and self-amplify if proteins (the proto-phenotypes) maintain the physical integrity of the template (the proto-genome) from which they are derived. This fundamental principle can be expressed as follows: The genome (in this case, an RNA) generates a phenotype (here, a protein) that contributes to the reproduction and stability of its originating genome by relaxing environment-dependent physicochemical constraints (Figure 1A). In this model, the stability of RNAs (the genome) and proteins (the phenotype) is interdependent, since by maintaining the stability of RNAs, proteins maintain their own sequence over time. In other words, a phenotype can only be reproduced across generations in a given environment if it maintains the stability of its originating genome.

In Section 5, it is shown that this principle captures the mechanisms by which cellular- and environment-dependent physicochemical parameters direct evolution. Indeed, it is shown that specific cellular- and environment-dependent physicochemical parameters exert constraints on some nucleic and amino acid polymers, which trigger a specific cellular response through the regulation of the expression of a selected set of genes. If the resulting cellular activities (the phenotype) relax the initiating constraints (i.e., return to equilibrium), this corresponds to physiological adaptation (Figure 1B, Path 1). Otherwise, it is shown that physicochemical constraints persist and challenge the physical integrity of nucleic and amino acid polymers inducing more-or-less direct mutations in the corresponding genome locations. This process, corresponding to genetic adaptation, only stops when new sequences directly or indirectly relax the initiating constraints (Figure 1B, Path 2). Therefore, cell activity (or physiological adaptation) and evolution (genetic adaptation) are based on the same principle, that is, the phenotype is derived from a genome in response to physicochemical constraints and relaxes the initial constraints. If the phenotype is adapted to a given environment (i.e., a set of physicochemical constraints), the genome is stable and will be reproduced “identically” (see Discussion). If the phenotype is unsuitable, the genome is unstable at specific locations (when directly or indirectly challenged by environmental fluctuations) and will be modified until a phenotype is generated that relaxes the initial constraints (i.e., that ensures the stability of the challenged genomic locations). 

If the main role of a phenotype is to maintain the stability (see Appendix A) of its originating genome, the phenotype (as the sum of all cellular activities) in turn creates physicochemical constraints on the genome. For example, the metabolic activities of cells produce a diversity of molecules (e.g., reactive oxygen species) that can interact with DNA and induce mutations. This means that a genome can generate a phenotype that can, in turn, generate physicochemical constraints on its originating genome (Figure 1C). In Section 6, this feedback loop is illustrated by showing that UV-radiation triggered the emergence of photosynthesis, which then triggered the emergence of cell respirationfollowed by eukaryogenesis. In addition, it is shown that this principle helps to explain the emergence of epigenetic modifications, multicellular organisms, and germline cells. Finally, it is highlighted that the interplay between the genome and the phenotype described above at the RNA/protein level is still operating in multicellular organisms when considering that the phenotype generated from the germline cell DNA corresponds to the production of somatic cells whose function is to protect the originating genome (i.e., the germline cell DNA) from environment-dependent physicochemical constraints.

It is concluded that the activities and evolution of living organisms are governed by the same physicochemical rule which is in disagreement with the current framework of evolutionary theory. Indeed, according to the current framework there is no direct relationship between physiological and genetic adaptation since physiological adaptation is based on physicochemical principles of homeostasis, whereas genetic adaptation would be fueled by random mutations (Figure 1D, left panel). In contrast, in the model proposed in this article, genetic adaptation is the consequence of physiological adaptation that takes place as long as fluctuations in environment-dependent physicochemical parameters do not exceed a physiological range. Above this physiological threshold, the integrity of nucleic and amino acid polymers, in particular DNA, is challenged leading to targeted mutations until the emergence of a phenotype that maintains the integrity of its originating DNA with regards to environmental constraints (Figure 1D, right panel).

It must be emphasized that rejecting the concepts of “random mutations” and “natural selection” as evolutionary “driving forces” does not at all exclude the possibility that these processes contribute to evolution, for the simple reasons that (i) all physicochemical (and therefore, biological) processes are stochastic; thus, mutational processes are probabilistic rather than deterministic and (ii) a living organism is necessarily adapted to its environment; therefore, natural selection constantly operates as a filter. In this light, the aim of this manuscript is not to define evolutionary driving forces that would explain all the diversity of living organisms. Instead, the evolutionary physicochemical driving forces, depicted here, should generate the fundamental shape of biological objects whose diversity could well depend on other phenomena as well, such as plasticity and natural selection. The depicted evolutionary driving forces allow us to describe evolution of living beings similarly to geological driving forces, in the way that plate tectonics describe earth evolution. Indeed, plate tectonics corresponding to physicochemical processes in the deep and superficial layers of the earth is a driving force explaining the formation of mountains and plateaus, but it does not explain the diversity of landscapes resulting from multiple contingent phenomena (e.g., wind, rain). This notion is further discussed in Section 6.

## 3. Environment-Dependent Physicochemical Constraints on Nucleic and Amino Acid Polymer Composition

Nucleic and amino acid polymers have emergent physicochemical properties (e.g., solubility, folding, and stability) that depend on intrinsic parameters relying on their composition and extrinsic parameters (e.g., temperature). The aim of this part is to show that cellular- and environment-dependent physicochemical parameters constrain the composition of cellular nucleic and amino acid polymers. In the next part, it is shown that this principle has consequences on the way evolution proceeds.

### 3.1. Physicochemical Constraints on Protein Composition

The amino acid composition of proteins is constrained by intrinsic parameters impacting on protein solubility, folding, and stability, for example, proteins with too many hydrophilic amino acids tend to unfold, whereas proteins with too many hydrophobic amino acids tend to aggregate [19,20]. Extrinsic physicochemical parameters, such as temperature or cellular and environment chemical composition, also constrain the amino acid composition of proteins, notably owing to the chemical modifications of amino acid side chains. Indeed, protein amino acids can react with and be modified by various chemical compounds, for example, lipids, sugars, and reactive oxygen species (ROS) [21]. Protein chemical modifications (see Appendix A) (i) are spontaneously or enzymatically generated, (ii) change the physicochemical properties of the modified amino acids, and (iii) contribute to cellular regulatory processes or induce protein mis-folding and aggregation [21]. For example, amino acid oxidation in proteins plays a role in many cellular processes, yet high rates of it induces protein damage and aggregation [21]. In agreement with the fact that biochemical compounds constrain the amino acid composition of proteins, organisms growing under different levels of salt or oxygen, or with different metabolic activities, produce proteins with different amino acid composition biases [20,22] (see Appendix A). Finally, the amino acid composition of proteins depends on the environmental supply of key elements that are required for the biogenesis of amino acids. For example, organisms growing in nitrogen- or sulphur-poor environments produce proteins that contain low amounts of nitrogen-rich amino acids (e.g., arginine) or sulphur-containing amino acids (e.g., cysteine), respectively [23,24]. In summary, the amino acid composition of proteins obeys physicochemical laws and depends on cellular- and environment-physicochemical parameters.

### 3.2. Physicochemical Constraints on Nucleic Acid Polymer Composition

RNA and DNA molecules are polymers composed of four monomers or nucleotides that interact with each other when they follow each other in a sequence (i.e., stacking interactions) or when they face each other in two different strands (i.e., base-pairing interactions). These interactions have consequences on the structural and physicochemical properties of nucleic acid polymers. G:C pairs form stronger base-pair interactions than A:T pairs, purine-purine dinucleotides (e.g., GpA) form strong stacking interactions, and GC dinucleotides form polymorphic structures [25,26,27]. As a consequence, GC- and purine-rich polymers are thermodynamically stable, and the genome of thermophilic organisms is enriched in GC or purine nucleotides [25,28]. Nucleotide composition also determines DNA mechanical properties (e.g., flexibility and bendability), contributing to its cellular functions and its resistance to torsional stresses (Figure 2A,B, see Section A.1). For example, increasing the GC content increases the B-DNA-to-Z-DNA conformational transition, DNA bendability, and DNA resistance to torsional stresses; accordingly, highly transcribed genes are GC-rich [26,29,30,31].

Similar to proteins, DNA and RNA molecules undergo dozens of spontaneous or enzyme-dependent chemical modifications (including oxidation, methylation, deamination, alkylation, and glycation) that (i) affect the physicochemical properties of nucleic acid polymers and (ii) contribute to regulatory processes but can also induce damages [32,33,34] (Figure 2C). For example, DNA oxidation not only plays a role in gene expression regulation but also increases DNA damages and mutations [32,33]. The reason why DNA chemical modifications can generate mutations during replication is because a chemically modified nucleotide can “mimic” another nucleotide. For instance, an oxidized guanine can base pair with an adenine rather than a cytosine, which can result in mutations during replication [32,33]. As for proteins, environmental physicochemical parameters (temperature and chemical composition) constrain the nucleotide composition of genomes and this can be observed in thermophilic, halophilic, acidophilic, aerobic, and radiation-exposed organisms [28,35]. Finally, the amount of DNA per cell (genome size and ploidy) is constrained by the environmental availability of phosphorus, and the genomes of organisms growing in a nitrogen-poor environment are enriched in A:T pairs, which require seven nitrogens instead of the eight used in G:C pairs [36,37,38].

In summary, the properties of nucleic and amino acid polymers depend on intrinsic parameters, and these polymers can undergo reversible structural and chemical modifications triggered by extrinsic physicochemical constraints. Above a certain threshold, these modifications can challenge their integrity (Figure 2A–D) and raises a major issue for coding sequences, which are under both nucleic acid-related and protein-related constraints (Figure 2E).

### 3.3. Interdependency between the Physicochemical Properties of Nucleic Acid Polymers and their Cognate Amino Acid Polymers

Coding sequences accommodate different constraints, by not only the encoded protein sequence but also cellular processes such as chromatin organization (e.g., nucleosome positioning), transcription (e.g., DNA flexibility), RNA folding, splicing, and RNA–RNA or RNA–protein interactions [39,40,41]. It has been assumed until now that variations of the third nucleotide of codons (the “wobble” position, which is usually the only variable between all codons encoding an amino acid) solves nucleic acid–related constraints without affecting the encoded amino acids [39]. However, this assumption is challenged by the diversity of the constraints described above, and by the fact that the third nucleotide of codons changes the thermodynamic property of codon–anticodon interactions, with consequences on translation fidelity, speed, and cotranslational protein folding [39,42]. Accommodation of different constraints in coding sequences also relies on the fact that the genetic code is not randomly organized, as amino acids that share physicochemical properties (e.g., hydropathy) correspond to codons with a similar nucleotide composition bias. For example, hydrophilic or hydrophobic amino acids correspond to A- or T-rich codons, respectively, and small or large amino acids are encoded by GC-rich or GC-poor codons, respectively [43,44]. Consequently, two codons with only one nucleotide difference (either at the first or third position) can encode either the same amino acid or different amino acids with similar physicochemical properties. 

However, the organization of the genetic code implies that the nucleotide composition bias of a nucleic acid polymer affects the physicochemical properties of the encoded protein and, conversely, that the amino acid composition bias in a protein affects the physicochemical properties of the cognate nucleic acid polymer. For example, compact proteins comprising small amino acids correspond to GC-rich coding sequences (as small amino acids correspond to GC-rich codons), while proteins comprising hydrophobic regions (i.e., containing stretches of hydrophobic amino acids) correspond to T-rich coding sequences (as hydrophobic amino acids correspond to T-rich codons) [30,45,46]. Along the same lines, the global nucleotide composition bias of a genome (e.g., GC content) is associated with a global amino acid composition bias of the encoded proteome [47,48].

Supporting the notion that constraints on the physicochemical properties of one kind of polymer affects composition biases of the cognate polymers are the following observations: (i) Nucleosome positioning leaves a footprint in protein sequences, (ii) splicing sites and splicing factor binding motifs constrain the amino acid composition of peptides encoded by splicing-regulated exons, and (iii) mRNA secondary structures depending on base complementarity have consequences on the secondary structures of the encoded protein [40,49,50,51]. Conversely, protein secondary structures leave a footprint in the nucleotide composition bias of coding sequences, for example, amino acids that favor alpha-helices and beta-sheets correspond to codons ending with purines and pyrimidines, respectively [52]. Along the same line, alternations of hydrophobic and hydrophilic amino acids in amphipathic alpha-helices that rely on a periodicity of ~3.5 amino acids correspond to a specific detectable ~10 bp periodicity in DNA, with consequences for the helical pitch of nucleosome-wrapped DNA [53]. Finally, purine enrichment in coding sequences is determined by protein-related physicochemical constraints, such as solubility and folding [54].

In summary, the common use of letters to symbolize biopolymers (such as DNA) obscures their physical nature which implies that their composition is constrained by environmental and cellular physicochemical parameters. This raises concerns regarding coding sequences that are constrained directly and indirectly by nucleic acid-related and protein-related parameters. While the organization of the genetic code “buffers” these constraints, nucleotide and amino acid composition biases affect each other above a certain threshold. As nucleotide or amino acid composition biases determine the physicochemical properties of polymers, the composition-dependent physicochemical properties of these interdependent polymers must be adapted to the same fundamental physicochemical constraints (Figure 2E), as is shown in the next Section.

## 4. Molecular Origin of Life and Evolution of the Genetic Code: Defining Evolutionary Driving Forces

Life relies on the interdependency between nucleic and amino acid polymers, since the biogenesis of proteins requires a nucleic acid polymer as a template (RNA), whereas biogenesis of nucleic acid polymers requires proteins. In addition, nucleic and amino acid polymers are in “competition” with each other, as their biogenesis requires the same elements (e.g., nitrogen) and requires the same template, i.e., single-stranded RNA (ssRNA) before the emergence of DNA. The aim of this section is to propose that these fundamental principles (interdependency and competition) constrain the genetic code evolution to match the fundamental physicochemical properties of both polymers. In other words, specific codons correspond to specific amino acids because their presence in nucleic acids and cognate proteins allows both polymers to deal with the same fundamental physicochemical parameters. This Part (i) defines the physicochemical forces driving evolution from the molecular origin of life and (ii) describes the bidirectional interplay in terms of stability between genomes and phenotypes.

### 4.1. Molecular Origin of Life: Interdependency between RNAs and Proteins

While there is an ongoing debate about whether the origin of life started with only RNAs (“RNA world”) or with RNAs and peptides (“RNP world”), there is a consensus that the activity of ribozymes (e.g., RNAs that catalyze nucleic acid polymerization) has been enhanced at some points of evolution by their interactions with amino acids or with randomly generated small peptides that could have also stabilized RNAs, which would otherwise be rapidly degraded by hydrolysis (Figure 3A) [55,56,57]. For example, randomly generated peptides composed of abiogenetically produced amino acids, such as Gly and Asp, can increase the efficiency of replicating ribozymes, as these amino acids play a very important role in the AsnAlaAspPheAspGlyAsp (NADPDGD) peptides found in all polymerases. In particular, binding of these amino acids to the catalytic metal ion Mg^2+^ could have enhanced polymerization and protected RNA from Mg^2+^-dependent hydrolysis [58]. Coincidentally, a primeval genetic code corresponding to GC-rich and RNY codons (e.g., GGC, GCC, GAC, and GTC) has been proposed, because these codons are the most frequent in coding sequences and correspond to the most metabolically simple amino acids (Gly, Asp, Ala, and Val) that are sufficient to produce stable, folded, and functional proteins [56,59]. A positive feedback loop between nucleic and amino acid polymers could have been initiated by primeval GC-rich RNAs, by first interacting with randomly generated peptides (e.g., made of Gly and Asp), which would then favor the production of more complex peptides through a primeval GC-rich genetic code.

However, such a cooperation would be inefficient if RNA and protein polymerization were physically uncoupled, for two main reasons. First, after replication, the ssRNA templates give rise to stable double-stranded RNAs (dsRNAs) which can decrease the rate of other rounds of replication, as well as that of protein synthesis that requires ssRNA templates [60,61]. Secondly, freely diffusible RNAs and proteins limit their cooperation, as freely diffusible peptides generated from an RNA template can stabilize and enhance the replication of potential “parasitic” or mutated RNA replicators [62,63]. Simulation and in vitro selection experiments demonstrate that ribozyme-dependent replication cycles rapidly end when molecules freely diffuse, due to the appearance of mutated RNAs, which become smaller, and therefore are more rapidly amplified while simultaneously losing their enzymatic activity [62,63]. Proto-cell compartmentalization and physical coupling between replication (i.e., RNA production in an RNP world) and amino acid polymerization (i.e., protein production) could solve these two main issues [64,65,66,67,68]. First, interactions between the nascent RNA and the nascent peptide could decrease the formation of stable dsRNAs while protecting ssRNAs from degradation. This is observed in modern prokaryotes or eukaryotes. Protein binding to nascent RNAs during cotranscriptional and translation or co-RNA processing prevents RNAs from interacting with the DNA template and increases transcription [69,70,71,72] (Figure 3B). Secondly, the physical proximity between replication and translation could increase the probability that the neo-synthetized proteins “protect” and enhance replication of its originating RNAs, which would have facilitated the increase in replication and translation fidelity [64,65,66,67,68].

While the physical coupling between RNA and protein polymerization seems to be a “sophisticated” molecular process, several authors have proposed straightforward models [65,66,67,68,73]. For example, it has been proposed that amino acylated trinucleotides corresponding to proto-tRNAs (tRNA ancestors) composed of three nucleotides (the proto-anticodons) and bound by one amino acid could have been used simultaneously for replication and translation. The three anti-codon nucleotides would have been used as building blocks during replication, while the attached amino acids would have chemically enhanced the triplet polymerization and been used as building blocks for protein production [74,75] (Figure 3C). Of note, (i) the use of tri-nucleotides rather than mono-nucleotides increases the efficiency and fidelity of replication by ribozymes [76]; (ii) phylogeny analyses suggest that tRNAs originated in replication [77]; and (iii) the coupling between transcription (i.e., RNA biogenesis) and translation is still operating in prokaryotes, and many features are shared by transcription and translation in modern cells [69,70,71,72] (Section A.2).

To summarize, life likely emerged from the cooperation between nucleic and amino acid polymers. The interdependency between RNAs and proteins defines the main evolutionary driving force. A genome (e.g., RNAs) generates a phenotype (i.e., proteins) that protects from environment physicochemical constraints its originating genome that can be replicated (Figure 1A). Therefore, the stability of the genome and the phenotype across generations is interdependent. This principle can help to understand the evolution of the genetic code. 

### 4.2. Evolution of the Genetic Code: Co-Adaptation of Nucleic Acid Polymers and their Encoded Proteins to the Same Fundamental Physicochemical Parameters

In a proto-cell without complex compartments and protein-dependent compensatory mechanisms, nucleic acids and proteins are exposed to the same physicochemical parameters (e.g., chemical compounds). As both polymers depend on each other, their composition and the processes that lead to their biogenesis need to satisfy the same physicochemical constraints. Supporting this model, GC- or purine-composition biases increase the thermostability of nucleic acid polymers and, in turn, correspond to codons of amino acids that increase protein thermostability [28,46,78].

Along the same line, the organization of the genetic code allows nucleic and amino acid polymers to co-adapt to the bioavailability of nitrogen. Indeed, A:T pairs require fewer nitrogen atoms than G:C pairs (7 vs. 8, respectively) and correspond to amino acids that also require fewer nitrogen atoms [23,36,37]. Accordingly, plant genomes and proteomes are AT-rich and contain amino acids that require fewer nitrogen atoms as compared with animal genomes and proteomes. This fits well with the fact that nitrogen sources are limited for plants, while animals can access organic sources of nitrogen [23,36,37]. Of note, in eukaryotic genes containing introns, exons and introns have the same composition bias [79]. Thus, AT-rich genes (N-poor) produce AT-rich mRNAs that code for N-poor proteins as compared with GC-rich genes.

Oxygen is highly toxic, and oxygen derivatives (e.g., ROS) can damage nucleic and amino acid polymers and induce their cleavage or aggregation [21,32,33]. Stepwise increases of oxygen in the biosphere over time (see Section 6 for more details) could have given the impulse for the late incorporation into the genetic code of amino acids that can act as “ROS scavengers” (including Trp, Tyr, Met, Cys, and His), and thereby protect biopolymers from oxidative damages [80]. Interestingly, the most frequent mutations in ROS-producing cancer cells affect Arg codons (and in particular, the CGN codons), with mutations producing codons for Cys (TGY), Trp (TGG), stop codons (TGA), and His (CAR) [81,82,83,84,85]. It has been proposed that these mutations (i) are induced by ROS-mediated deamination of (methyl) cytosine, leading to C > T mutations, or incorporation of oxidized guanine during replication, leading to G > A mutations and (ii) protect cancer cells from the high levels of ROS by increasing the global antioxidant capacity of the cancer proteome [81,82,83,84,85]. Interestingly, CGN codons (that encode Arg) seem to be particularly sensitive to oxidation because of their physicochemical properties [81,86]. Furthermore, (methyl)cytosine deamination produces CG > TA mutations, and thereby can increase the nitrogen availability required by proliferative cancer cells, as it reduces the nitrogen-richer C:G pairs in favor of the nitrogen-poorer T:A pairs, as well as the GC-rich sequences that contain codons (e.g., CGN) corresponding to nitrogen-rich amino acids (e.g., Arg) [81]. Therefore, ROS-induced cytosine deamination would simultaneously save nitrogen atoms (at both genome and proteome levels) and protect cells from ROS. Although speculative, one possibility is that the assignment of “ROS scavenger” amino acids to codons that originate from the oxidation of Arg codons generates antioxidant proteins from oxidized nucleic acids. Supporting such a possibility, assignment of Met to the ATA codon (in addition to the ATG codon) in most animal mitochondria lineages explains the high frequency of Met in mitochondria-encoded respiratory chain complexes, and it represents an adaption to high ROS level in mitochondria [87].

If physicochemical constraints shaped the genetic code, it is very likely that the universal genetic code is the result of horizontal transfers of “code fragments”, as proposed by several authors [57,88]. For example, protocells that are present in extreme environments (e.g., hot or cold, nitrogen rich or poor, oxygen rich or poor) could have developed “code fragments” adapted to their specific environment. At the frontiers of these habitats where physicochemical parameters fluctuate, genetic horizontal transfers between cells using different “code fragments” could have led to the emergence of the modern genetic code (Figure 3D).

To summarize, the organization of genetic code has been constrained to match general physicochemical properties of the interdependent nucleic and amino acid polymers. This means that RNAs and proteins could only coevolve if RNAs and their cognate proteins are adapted to the same physicochemical constraints (temperature, N availability, etc.). Moreover, cooperation between RNAs and proteins must have required that proteins preferentially interact with their encoding RNAs in order to (i) protect their cognate RNA from damage and (ii) limit protein diffusion, and thus limit their use by parasitic RNAs (see above for description of the coupling between nucleic and amino acid polymerization). Supporting this possibility, RNA binding proteins bind preferentially to mRNA coding sequences that have similar nucleotide composition biases as their own encoding mRNAs [89,90,91]. In other words, and as an extension of the stereochemical hypothesis, the genetic code could have been shaped over evolutionary time to allow the interactions between proteins and their cognate mRNAs, making their cooperation possible by protecting and ensuring the stability of each other (Section A.2). Somehow, self-assembly of proteins with their encoding RNA in viral capsids could reflect this primitive feature and function of proteins (i.e., proto-phenotype) to protect their encoding RNAs (i.e., proto-genome) [92]. 

The intimate cooperation and interdependency between nucleic and amino acid polymers imply that the genetic code has been shaped over evolutionary time to ensure that nucleic acid polymers and their cognate proteins share more physicochemical and structural properties than previously anticipated (Figure 2E and Figure 3E). In Section 5, the consequences of these sharing properties are shown in terms of evolution, after having described the interplay between the cell metabolism and nucleic and amino acid polymers.

### 4.3. Feedforward and Feedback Loops between Gene Products and their Products (i.e., Metabolites)

Nucleic and amino acid polymers depend on the availability of molecules containing elements, such as nitrogen (see Section 3). This dependency would have favored the emergence of polymers with metabolic activities that modify environmental resources and allow elements to be incorporated into amino acids and nucleotides. In an RNP world, the proto-cell phenotype would be a positive feedback loop between biosynthetic pathways (i.e., metabolism in modern cells) and polymerization of RNAs and proteins (i.e., the gene expression process in modern cells). Polymer production depends on biosynthetic pathways that in turn depend on polymerization products (Figure 3F). Supporting this interplay between gene expression and biosynthetic pathways, the genetic code likely coevolved with amino acid biosynthetic pathways, as codons starting with A, C, U, or G correspond to amino acids synthetized from oxaloacetate, alpha-ketoglutarate, pyruvate, or from the reductive amination alpha-keto acid, respectively [56,93,94,95]. A possibility is that the simple amino acids or metabolites that covalently attached to polynucleotides (e.g., proto-tRNAs made of three nucleotides) could have been chemically transformed to give rise to more complex amino acids [95]. Another possibility is that chemical modifications of simple amino acids occurred after their incorporation into proteins, and that chemically modified amino acids (i.e., complex amino acids) were, thus, available after protein hydrolysis. Then, the complex amino acids would have been incorporated into the genetic code.

In this context, it must be underscored that the chemical modifications of proteins and nucleic acids establish a direct bridge between cell metabolic activities and gene expression and gene product functions (i.e., the cellular physiological adaptation). Indeed, chemical modifications of nucleic and amino acid polymers (e.g., post-translational modifications, DNA, or RNA methylation) that change their physicochemical properties also affect their cellular activities, and these chemical modifications depend on the cell metabolic activities. For example, DNA, RNA, and protein oxidation depend on cellular oxidative metabolism, while DNA, RNA, and protein methylation depend on the production of *S*-adenosyl methionine (SAM), the universal methyl-group donor produced in the one-carbon cycle [96,97,98]. 

In summary, the genetic code could only coevolve with metabolic pathways that provide amino acids required for protein synthesis. The necessity of this coevolutionary process comes from the fact that a cell’s autonomy is based on the biogenesis of biopolymers (RNAs and proteins), which allows the transformation of molecules from the environment into metabolites (nucleotides and amino acids) that are required for biopolymer synthesis (Figure 3F,G). This concept corresponds to the notion of autopoiesis proposed by H. Maturana and F. Varela [99,100]. Indeed, autopoiesis was defined as the property of a system to produce itself, and therefore to be autonomous by maintaining its organization despite any change in its components. In addition, the interplay between metabolites and biopolymers (RNAs and proteins) (Figure 3G) shares features with hypercycles described by Eigen and Schuster [63,101,102], Hypercycles that correspond to an organization model of molecules connected in a cyclic and autocatalytic manner have been proposed to play a major role in self-organization and self-reproduction, increasing the fidelity of interdependent processes while limiting the reproduction of parasitic elements.

Very importantly, metabolites are not only necessary for polymer biogenesis but can also react with them, leading to polymer biochemical modifications. These biochemical modifications (e.g., epigenetic modifications of DNA and RNAs, or post-translational modifications of proteins) change the physicochemical properties of the polymers and play a major role in the physiological adaptation of cells in response to environmental variations; however, at the same time, biochemical modifications can trigger polymer damages (e.g., protein aggregates and DNA mutations). One hypothesis is that the composition of biopolymers in cells is adapted to the nature of the metabolites that the cells produce (see Section 3). Thus, within a physiological range of metabolite concentration, metabolite-dependent biochemical modifications of polymers contribute to cellular physiological adaptation; however, beyond a certain physiological threshold, biochemical modifications directly or indirectly induce mutations (Figure 2C and Figure 3H). This suggests that biochemical modifications of nucleic and amino acid polymers establish a continuum between physiological and genetic adaptation.

## 5. Continuum between Physiological and Genetic Adaptation

The aim of this section is to show that environmental fluctuations induce physical and chemical constraints on nucleic and amino acid polymers, which trigger the cellular physiological adaptation that maintains cellular homeostasis (Figure 1B, Path 1). However, if the cellular response to environmental fluctuations does not return to equilibrium, constraints persist and can challenge the physical and chemical integrity of DNA, leading to DNA damage, mutations, and genetic variations. This process only ends when mutated sequences allow the direct or indirect relaxation of the initial constraints (Figure 1B, Path 2). First, how physical and chemical constraints originating from the environment or from cellular activities increase the probability of inducing directed and adaptive mutations, thus, “directing” genetic adaptation is shown and, then, the role of RNAs in genetic adaptation, including in multicellular organisms is discussed.

### 5.1. Genetic Adaptation Directed by Transcription: Transcription-Replication Conflicts

Several authors have already proposed that cellular stresses (i.e., environmental fluctuations above a physiological range) induce “adaptive mutations” (i.e., mutations that occur at a high rate) or “directed mutations” (i.e., mutations occurring at specific genomic locations) [103,104,105]. A possible underlying mechanism is that stress-directed transcriptional activation of specific loci (as part of the cellular physiological adaptation) increases the probability of mutations occurring within these loci because transcription induces mechanical stresses (e.g., formation of supercoiling) that challenge the physical integrity of transcribed DNA [12,34,104,105,106,107]. Next, how this straightforward principle establishes a continuum between physiological and genetic adaption is described below.

Highly transcribed genes are enriched in GC nucleotides that can “absorb” mechanical stresses by favoring the B- to Z-DNA transition, owing to the physical properties of both base-pairing and base-stacking interactions between G and C nucleotides (see Section 3). Critically, this GC enrichment is explained by several transcriptional-dependent mutational biases (see Appendix A), which result, in part, from conflicts between transcription and replication [107,108]. Indeed, while the act of transcription increases DNA accessibility to DNA polymerases, the simultaneous transcription and replication of a locus creates conflictual physical stresses leading to DNA breaks [107,108,109]. DNA breaks can be repaired by heteroduplex DNA recombination, which is a process known to favor G:C more than A:T base pairs [110,111,112,113]. This phenomenon, in part, relies on the fact that a T in a mismatched base-pair (T:G or T:C) within heteroduplex DNA can spontaneously flip out of the dsDNA, increasing the probability of its removal [114]. In addition, transcription-replication conflicts have been shown to induce adenine deamination, giving rise to hypoxanthine, a nucleotide that mimics guanine and leads to A:T > G:C mutations during replication [115]. Different mutational biases also occur in early and late replicating regions [7]. For example, the accumulation of free oxidized-dGTP before replication can result in its incorporation in place of Ts (as oxidized-dGTP mimics adenine), leading to A:T > G:C mutations during the early phase of replication [116]. Additionally, as DNA cytosine methylation is associated with transcription repression, heavily methylated regions correspond to late replicating regions; these regions could more frequently undergo C:G > T:A mutations because of the high rate of spontaneous deamination of methylcytosine [117]. Replication-dependent mutational bias can also be due to decreases during the cell cycle of concentrations of free dGTPs and dCTPs (as producing these nucleotides requires more energy than producing dATP and dTTP), which leads to a higher incorporation rate of A or T nucleotides in late-replicating regions [118]. Therefore, mutational biases associated with replication timing and replication-transcription conflicts could increase the GC- and AT-content in early and late replicating regions, respectively.

While the GC-content of loci can increase as a consequence of DNA breaks resulting from transcription-replication conflicts, then, this increase exerts positive feedback loops by (i) synchronizing transcription and replication, (ii) increasing local DNA stability during transcription or replication, and (iii) increasing the transcription activity of modified loci as well as many downstream steps of the gene expression process [27,31,41,119]. Indeed, the mechanical properties of GC-rich DNA regions favors transcription efficiency (but not elongation speed) and avoids nascent RNAs from interacting with the DNA template due to the formation of stable secondary structures in the nascent GC-rich RNAs [69,70]. High GC-rich content can also increase the local rate of recombination and deletion, leading eukaryotic GC-rich genes to bear smaller introns (as compared with AT-rich genes) [29,30,112,120]. Of note, small GC-rich introns are more efficiently spliced, likely because of intronic RNA secondary structures [79,121]. In addition, high GC content in RNAs increases the efficiency and fidelity of translation by smoothing translation elongation and favoring cotranslational protein folding [51,122,123,124]. Finally, as the genetic code is not random, increasing gene GC content leads to the biogenesis of proteins with small amino acids, which in turn leads to decreases in: (i) protein volume, (ii) concentration-dependent aggregate formation, and (iii) the energetic cost of protein production [30,125,126,127]. Therefore, the “over-stimulation” of the transcriptional activity of some genes under sustained stresses could result in (i) replication-dependent mutational bias; (ii) increases in the GC-content of stress-induced genes; and (iii) decreases in the local transcription-dependent DNA instability, while improving gene product biogenesis at multiple levels (Figure 4A). Another genetic process that increases the biogenesis of specific gene products under stress situations is gene duplication, which is also linked to transcription-replication conflicts [12,128]. For example, stress-induced promoter activity can stimulate gene duplication by destabilizing stalled replication forks [128]. 

These observations, therefore, support a model in which environmental fluctuations induce physical constraints on DNA during transcription, leading to the biogenesis of RNAs and proteins, and, then, leading to re-establishing equilibrium through the cellular physiological adaptation (Figure 4A, 1). However, if the constraints persist, sustained transcriptional activation can result in transcription-replication conflicts resulting in GC-mutational biased or gene duplication, which could relax the initiating constraints by increasing gene product levels (Figure 4A, 2).

### 5.2. Genetic Adaptation Directed by Transcription: Role of ssDNA Formation and DNA Folding

Transcription-dependent chromatin relaxation and ssDNA formation increase the accessibility of transcribed DNA regions to mutational agents or so-called “transcription-associated mutations” [5,12,34,105,106,107,129]. For example, ROS-mediated oxidation of cytosines and methylcytosines is enhanced in ssDNA, which can lead to C > T mutations during replication [34]. As a consequence, CGN codons (corresponding to Arg) of transcriptionally activated genes under ROS exposure have a higher probability to be mutated into TGG, TGA, and TGY codons (corresponding to Trp, stop, and Cys codons, respectively), allowing the synthesis of proteins containing amino acids that protect cells from oxidation (see Section 4 and Figure 4B).

Remarkably, a genome-wide analysis of mutations occurring in organisms growing under different environmental constraints (e.g., different metabolic resources or temperatures) shows that each challenging condition is associated with a specific mutational bias [130,131,132,133,134] (see Appendix A). This is in agreement with the fact that each mutagenic agent (i) affects sequences with specific physicochemical properties and (ii) induces nucleotide modifications toward a particular pattern, as has now been well established in cancer genetics [135]. Of note, in addition to ROS-associated mutational signatures in cancers (see Section 4), mutational bias induced by high intracellular pH has recently been shown to favor Arg (CGY) > His (CAY) mutations that confer pH-regulated protein functions [136]. It would be very interesting to systematically characterize the relationship between (i) the physicochemical properties of mutagenic agents as well as of their most affected sequences and (ii) the nature of the induced mutations and the resulting physicochemical consequences at the nucleotide and amino acid levels [8]. Each mutagenic agent could trigger a specific mutation bias that more or less directly relaxes the initial physicochemical constraints (Figure 1B, 2).

It is of particular interest that transcription-dependent ssDNA formation also increases the probability of insertion of repeated elements, such as transposons and retrotransposons, a process known to play a major role in the genomic plasticity under sustained stresses [137,138,139,140,141]. It has been proposed that cellular stresses leading to a global chromatin relaxation could, on the one hand, de-repress (retro) transposon activity and, on the other hand, increase the likelihood of their insertion in specific stress-transcriptionally activated genes [137,138,139,140,141]. Insertion of (retro) transposons in stress-activated genes can influence gene expression at multiple levels, particularly by playing a role in the spatial genome organization. Indeed, it is now well recognized that regulation of gene transcription is based on the three-dimensional (3D) genome organization, which roughly corresponds to DNA folding (Section A.1). DNA folding plays a critical role in co-regulating genes by bringing them closer together in space [142,143,144,145,146]. Factors binding to repeated sequences dispersed in various genomic locations could facilitate 3D genome organization and promote co-regulation of repeated sequence-hosting genes [147,148] (Figure 4C). Spatial clustering of co-regulated genes could also increase the probability of their recombination, a process facilitated by the presence of repeated elements [149,150,151,152]. Recombination between transcriptionally co-regulated genomic regions can lead to the formation of new gene products but also can facilitate the expression coordination of costimulated genes (Figure 4C). The importance of gene position in genomes with respect to their regulation and cellular functions is clearly established in operons from bacteria and in the so-called topologically-associated domains (TADs) in eukaryotes [153,154,155,156]. An emerging theme is that the one-dimensional (1D) and 3D locations of genes play a major role in coordinating the expression of genes whose products are involved in the same cellular pathways (Section A.1). 

In summary, environment-dependent physicochemical constraints on DNA trigger cellular physiological adaptation and a continuum between physiological and genetic adaptation is established when environment-dependent physicochemical parameters above a physiological range challenge DNA physicochemical integrity.

### 5.3. Physiological Adaptation Facilitates Genetic Adaptation: Role of RNAs

DNA chemical modifications can contribute to physiological adaptation (e.g., the role of DNA methylation and oxidation in transcription regulation) and induce mutations during replication when a chemically modified nucleotide “mimics” another nucleotide (see Section 3). This mimicry process can also result in transcription “infidelity” by affecting the base-pair rules, thereby leading to biogenesis of RNAs with different sequences than those encoded by the DNA template [157,158,159]. This so-called “transcriptional mutagenesis” is more frequent than previously anticipated and could play a major role in both physiological and genetic adaptations. For example, chemical modifications in transcribed DNA can lead to the biogenesis of new RNAs and proteins, which could contribute to cell survival. As a consequence, only cells bearing the DNA chemical modifications can divide, and therefore having the DNA chemical modifications would increase the probability of giving rise to genetically adapted daughter cells [157,158,159,160,161,162] (Figure 4D). The same principle can be used to establish a direct link between genetic adaptation and epigenetic modifications (i.e., DNA or histone chemical modifications that impact gene expression), as environmental fluctuations induce epigenetic modifications of specific loci as part of the cells’ physiological adaptation. If transcription-dependent epigenetic modifications at specific loci increase cell survival, the surviving cells have a higher probability of undergoing mutations within these genes, as chromatin organization and epigenetic marks more or less directly impact both DNA damage and DNA repair [34,163,164].

Along the same line, RNAs produced from loci as part of the cellular physiological adaptation could, under certain circumstances, induce mutations in the loci they originated from. For example, the spatial proximity of co-regulated genes (see above) could promote the biogenesis of chimeric RNAs via a mechanism called trans-splicing, thereby “fusing” RNAs produced from two genes [165]. These chimeric RNAs can give rise to new proteins that could contribute to the survival of cells in a stressful situation. As RNAs can be used as a matrix during DNA break repair, surviving cells expressing chimeric RNAs could use these RNAs during the repair of loci broken by the stress-induced transcription, increasing the probability of recombination between specific loci [166,167]. 

RNAs can increase the likelihood of genetic variations in numerous ways [168,169]. Recently, spectra of molecular mechanisms have been described by which physicochemical constraints on RNAs and proteins, at the time of their synthesis, could trigger mutations in their originating genes through the biogenesis of RNA fragments [72,170,171]. Briefly, environmental fluctuations, on the one hand, induce the transcriptional activity of target genes, and thereby generate a greater amount of mRNAs and proteins, and, on the other hand, generate constraints on nascent RNAs and nascent proteins during transcription and translation (Figure 4E, 1 and 2). Perturbation of mRNA or protein synthesis leads to the biogenesis of RNA fragments, for example, mRNA cleavage occurs when the dynamics of ribosomes (e.g., as a consequence of nascent protein misfolding) along the mRNA template is disturbed [172,173]. RNA fragments generated during transcription or translation could then interact with their originating genomic regions and induce genomic instability and mutations in the targeted regions (Figure 4E, 3). Therefore, RNA-directed mutations could increase the likelihood of mutations occurring in specific loci when cells experience constraints during the biogenesis of specific RNAs or proteins.

In summary, environment-dependent physicochemical parameters trigger cellular physiological adaptation through changes of the cellular activities that leave traces or footprints on nucleic acid polymers through physical damages (e.g., DNA breaks), chemical modifications (e.g., DNA oxidation), and biogenesis of RNAs that can, next, target specific genomic locations. If the physiological adaptation allows a return to equilibrium, polymer modifications are temporary and reversible (Figure 1B, 1). If not, the footprints left on DNA (i.e., the “cell experience”) can have a more-or-less direct effect on replication, potentially leading to mutations at specific loci. This mutational process triggered by physicochemical constraints only stops when new sequences relax the initiating constraints (Figure 1B, 2). This principle is well suited to unicellular organisms in which the same DNA molecule is used as a template during (i) the physiological response to environmental fluctuations and (ii) replication and transmission across generations. Could such a principle apply to multicellular organisms?

### 5.4. Somatic Physiological Adaptation and Germline Genetic Adaptation: Role of RNAs

It has been proposed above that environment-dependent physicochemical constraints on DNA trigger a cellular physiological adaptation that can leave “marks” (e.g., DNA chemical modifications or RNAs), which, then, potentially result in mutations during replication. As the replicated-DNA that is transmitted across generations in multicellular organisms is no longer directly involved in physiological adaptation, do physicochemical constrains exerted on somatic cell phenotype induce modifications on germline cell DNA? 

To answer this question, first, it must be stressed that germline cells depend on and are exposed to the activities of somatic cells. For example, metabolic disorders are associated with the overproduction by somatic cells of small sugars or lipids that can react with and induce damages in the germline cell DNA. Furthermore, metabolic activity of somatic cells, for example under nutrient constraints of the parents, can induce epigenetic changes on specific loci in germline cell DNA with consequences on the development and activity of the offspring [174,175,176,177]. There is also a consensus regarding the fact that molecular exchanges between somatic and germline cells are more complex than previously anticipated. For example, somatic cells produce extracellular vesicles that (i) contain proteins, metabolites, and a diversity of small RNAs, for example, microRNAs, tRNA-derived small RNAs (tsRNAs) and (ii) are internalized by germline cells with consequences on the development and activity of offspring after fertilization [176,178,179,180,181,182] (Figure 4F, 1). These processes, collectively called “transgenerational epigenetic inheritance”, establish that a somatic cell’s “experiences” in the parent organism can be transmitted at the molecular level to germline cells, with consequences on the offspring’s phenotype [176,178,179,180,181,182,183]. Another process, called “genomic imprinting” that also depends on germline epigenetic marks and small RNAs leads to the selective expression of alleles transmitted from one of the two parents sometimes in an environment-dependent manner [184]. Epigenetic marks of one allele can also be transferred to the other allele, a process called “paramutation”, which is based on the biogenesis of biochemically modified small RNAs and which underlines the plasticity associated with epigenetic mechanisms [185,186]. All the processes described above maintain a diversity of alleles by transcriptionally activating or repressing some of them in the offspring, depending on the parental somatic cells’ experiences (Figure 4F, 1 and 2) and combined with meiosis, these processes could “purge” some deleterious alleles [187,188,189,190,191,192,193].

Indeed, the frequency of transmission of some alleles in offspring can vary depending on their parental origin. This mechanism, known as the “transmission ratio distortion”, is an exception to Mendel’s laws of equal segregation and seems to rely on a diversity of mechanisms, such as “selection” among chromosomes during meiosis (e.g., non-random crossovers during meiotic recombination), allele-dependent elimination of gametes, or selective elimination during early zygote development [191,192,193]. Although the underlying molecular mechanisms of transmission ratio distortion are not yet fully understood, such a process establishes that sexual reproduction allows the selective elimination of some alleles. One possibility is that genes targeted by somatic RNAs in germline cells have a lower probability to be transmitted in next generations depending on the parents’ experiences (Figure 4F, 3 and 4). Therefore, sexual reproduction allows genes to be turned on and off, and for some alleles to be eliminated across generations without the need of de novo mutations in germline cells.

In this context, several studies have shown that, although the occurrence of germline de novo mutations per generation is very low in some species, their distribution across genomes is biased [9,10,194,195]. For example, an association between de novo mutation occurrence, replication timing, transcription, and chromatin organization has been observed in germline DNA [9,10,194,195]. Furthermore, de novo “mutational clusters” corresponding to multiple de novo mutations in very close vicinity in a single individual, as well as “mutational hotspots” corresponding to de novo mutations occurring at the same location in several individuals, have been reported [194,195,196]. As the distribution of de novo mutations across genomes is biased, an important issue is to decipher whether their occurrence could depend on the somatic cell experiences. One possibility could rely on the fact that RNAs produced by somatic cells induce local and targeted epigenetic modifications in the germline DNA, which next induces more or less directly targeted de novo mutations because of the interplay between the chromatin environment and the local mutational rate (Figure 4G, blue arrows) [34,163,164,197,198]. Although very speculative, there is also the possibility that the DNA of somatic cells challenged by environment-dependent physicochemical parameters produces “parasite-mimicking RNAs” that could form DNA:RNA hybrids in their complementary loci in the germline DNA and locally trigger mutations (Figure 4G, red arrow). The following supports this possibility: (i) A convergence between RNA-containing extracellular vesicles and viral particles has been described; (ii) RNAs are widely used in all living organisms to cleave or mutate parasitic nucleic acids; and (iii) RNA:DNA hybrids can be genotoxic, for example, RNA:DNA hybrids can induce DNA adenine deamination [72,168,169,170,171,185,186,199,200,201,202,203,204]. It would be very interesting in the future to investigate whether some of the 150 chemical modifications of RNAs identified so far could trigger selective mutations in RNA:DNA hybrids [72,168,169,170,171,185,186,201,202,204].

To summarize, in unicellular organisms, the cell’s experience leaves footprints on specific DNA locations (e.g., breaks and chemical modifications) that can lead to local mutations during replication, thus, establishing a continuum between physiological and genetic adaptation. In multicellular organisms, somatic cells challenged by environment-dependent physicochemical parameters may not properly protect germline cell DNA from physicochemical constraints and could produce compounds (e.g., RNAs) that target specific locations of germline cell DNA. Section 6 provides a description explaining how these mutational processes, and the interplay between the genome and phenotype stability contribute to the emergence of more-or-less complex phenotypes, including those in multicellular organisms.

## 6. Interplay between Environment-Dependent and Cell-Dependent Physicochemical Constraints and the Emergence of Complex Phenotypes

On the basis of the relationship between RNAs and proteins at the origin of life, in Section 4, it is proposed that evolution relies on the fact that the phenotype maintains the physiochemical stability of its originating genome in a manner that depends on environmental, physicochemical constraints (Figure 1A). One of the objectives of Section 6 is to show how this principle leads to molecular innovations that allow the emergence of new physicochemical properties at higher scales of life organization, i.e., complex phenotypes that can next act on the initial constraints (Figure 5A, blue lines). Furthermore, the aim is to show that the phenotype (irrespective of the scale of organization) generates constraints on its own genome. In other words, environmental constraints induce molecular innovations that can also directly or indirectly generate constraints on their originating genome (Figure 5A, red lines), and therefore shows how this principle creates an evolutionary dynamic. Throughout Section 6, it is stressed that the mutational processes described in Section 5 are not deterministic but rather probabilistic, and therefore are the source of variability, and therefore diversity. Similarly, the fact that the emergence of certain properties under physicochemical constraints can be considered as “side effects” that contribute to life form diversity is highlighted.

### 6.1. From Molecular Innovations to Emergence of New Properties at Multiple Scales of Life Organization: Metabolic Activities and Cell Organization

The phenotype (i.e., the sum of all cellular activities) necessarily generate physicochemical constraints on its originating genome since numerous compounds are produced by the cell in response to environment-dependent physicochemical variations. The compounds can interact and react with nucleic and amino acid polymers with potentially deleterious effects on their stability, folding, or solubility [21,205] (Figure 5B and see Section 3). Therefore, cellular activities represent a source of physicochemical constraints on the originating genome. The emergence of photosynthesis, oxidative metabolism, and eukaryogenesis illustrate the evolutionary dynamic generated by the interplay between phenotypic and genomic stability. 

Before the emergence of the ozone layer, cells were exposed to strong solar UV radiations [206,207,208]. By inducing genomic instability, UV favored the emergence of genomes that produce pigments absorbing UV-damaging radiations, which would give these genomes a greater probability to be “accurately” reproduced [206,207,208] (Figure 5C, 1). Interestingly, it has been proposed that the molecular ancestors of photosynthetic light acceptors (e.g., chlorophyll) were pigments that protected nucleic and amino acid polymers from UV irradiation [206,207,208]. One possibility is that UV-absorbing pigments generated genomic instability because of the free dispersion of energy (heat) released from UV-absorbing pigments, favoring the emergence of pigment-interacting proteins containing photosynthetic reaction centers that could concentrate energy into complex molecules (e.g., sugars or “energy tanks”) by fixing CO_2_ [206,207,208] (Figure 5C, 2). This gave rise to photosynthesis that in turn generated new constraints, as photosynthesis produces oxygen, a highly toxic compound that damages nucleic and amino acid polymers. In this setting, the ancestors of the gene products and metabolites involved in cell respiration have been proposed to have originated from oxygen scavenger compounds or oxidases that did not conserve energy and that protected their originating genome from the rise of oxygen [209,210,211]. Therefore, the emergence of cellular respiration could have resulted from a “detoxification” process concentrating oxygen-derived energy in biosynthetic pathways, in the same way that photosynthesis emerged from energy concentration from UV radiations [209,210,211] (Figure 5C, 3). 

The rise of oxygen produced by photosynthetic cyanobacteria in the earth atmosphere could also have impelled anaerobic cells (e.g., archaeas) and aerobic cells (e.g., alphaproteobacteria) to cooperate, as aerobic cells could protect anaerobic cells by scavenging environmental oxygen and because both cell types exchanged intermediate metabolites [212,213,214]. This cooperation might have resulted in the internalization of aerobic bacteria (the ancestors of mitochondria) by anaerobic archaea, resulting in the emergence of eukaryotes [212,213,214]. However, as mitochondria, subsequently, generated intracellular toxicity by producing intracellular ROS, this detoxification would have favored the biogenesis of new cellular compounds [215,216]. In this setting, biogenesis of sterols that requires oxygen-dependent enzymes could have first played a role in oxygen detoxification [216,217,218]. In addition, these molecules have specific properties when incorporated into membranes that contribute to the development of eukaryotic intracellular membranes, such as the nuclear membrane, which could have initially protected intracellular polymers (e.g., DNA) from ROS [216,217,218,219]. Indeed, intracellular biomembranes can fold up into three-dimensional periodic arrangements (”cubic membranes”), representing antioxidant defense [220]. 

In summary, the step-by-step emergence of photosynthesis, oxidative metabolism, and eukaryogenesis over evolutionary time could have been triggered by extracellular (e.g., UV radiation) and intracellular (e.g., ROS) physicochemical constraints that destabilize genomes (i.e., induce genetic variations). This resulted in the emergence of new genomes that produced molecules that relaxed the initiating constraints (molecular innovations), therefore, stabilizing their originating genomes but, simultaneously, generating new constraints (Figure 5C). The emergence of intracellular membranes triggered by the increase of intracellular ROS also illustrates that molecular innovations in response to physicochemical constraints (e.g., sterol metabolism as a ROS detoxification process) supports the emergence of new properties at the upper level of life organization (e.g., cell organization based on intracellular membranes). At the cellular level, these properties (i.e., cell compartmentalization), then, contribute to reducing ROS-dependent genomic instability. Furthermore, the interplay between molecular innovations and the emergence of new properties at the (multi)cellular level is illustrated beow.

### 6.2. From Adaptation to a Diversity of Environment-Dependent Constraints, to Side Effects: Genome Organization, Epigenetics, and Multicellularity

The increase in intracellular ROS levels produced by mitochondria in eukaryotes could have been relevant to the origin of eukaryotic spliceosomal introns from group II introns found in archaea and bacteria, which are in fact mobile retroelements that use the combined activities of an autocatalytic RNA and an intron-encoded reverse transcriptase to propagate within genomes [221,222]. It has been proposed that retromobility of group II introns can be stimulated by oxidative stress and that the presence of introns could have contributed to “trapping” ROS in introns, thereby, decreasing the probability of nucleotide oxidation in coding exons [221,223,224,225]. Supporting this model, nucleotide oxidation increases the probability of GC > AT mutations, and the frequency of GC nucleotides is higher in exons than in introns [225,226]. This implies that exons have been protected from ROS, which could have been achieved by histones. Indeed, histones are preferentially found in GC-rich sequences because of the flexibility of the G–C stacking interactions (see Section 3), and they protect DNA from a variety of mutagenic stresses by (i) binding to and stabilizing dsDNA, (ii) compacting DNA, and (iii) providing a “shield” through their C-terminal tails that are rich in Arg and Lys, i.e., amino acids that are basic and positively-charged and can act as ROS scavenger [227,228,229,230,231]. Therefore, binding of histones to GC-rich exons can protect them from oxidation, while intronic GC > AT mutations induced by ROS would ultimately lead to exclude histones from introns. If introns contributed to maintaining the stability of exons, a side effect of intron invasion is that it increased the diversity of proteins produced by eukaryotic genomes through alternative splicing. Similarly, the diversity of genome-driving phenotypes could be a side effect of the emergence of histones.

Indeed, the cell physiological adaptation to environmental fluctuations relies on the biogenesis of gene products and metabolites. Nevertheless, all possible biochemical reactions cannot take place simultaneously in a cell as (i) each reaction depends on specific physicochemical conditions and (ii) the diversity of the generated biochemical products would be highly toxic, due to their ability to interact and react spontaneously with each other [21,205] (see Section 3). Therefore, a cell can physiologically adapt to a limited number of fluctuating physicochemical parameters, which pushes towards the cooperation between unicellular organisms performing complementary biochemical reactions [232] (Figure 5D).

In this context, histones could have allowed an increase in the diversity of metabolic activities encoded by a single genome [233]. Eukaryotic histones evolved by compacting DNA and by acting as a “chemical” shield, therefore, maintaining the stability of the genome that produces them (see above). Histone chemical modifications (i.e., epigenetic marks) could first have been triggered as chemical “shields” and played a role in maintaining DNA stability against chemical DNA “attacks” [234]. However, by affecting DNA accessibility, histone chemical modifications would not only have protected specific genes but also coordinated their activity depending on the intracellular chemical composition. Indeed, different epigenetic marks can protect different parts of the genome from different chemical compounds, while simultaneously adapting gene transcriptional activities with respect to these chemical compounds (Figure 5E). Two pieces of evidence support such a possibility. First, epigenetic marks are directly dependent on the cell metabolism, for example, methylation depends on SAM produced by the one carbon cycle, and demethylation relies on oxidation of methylated residues, and therefore on the cellular oxidative metabolism [96,97,98,233]. Secondly, histone chemical modifications either reduce or facilitate DNA access to RNA polymerases and to potentially genotoxic molecules [97,98,163,234,235]. By selectively protecting and regulating gene expression, histone chemical modifications contributed to the emergence of different cell types (i.e., multicellularity) containing the same genome but performing different metabolic activities (Figure 5E).

In summary, molecular innovations triggered by physicochemical constraints allow (directly or as side effects) the emergence of new properties at higher scales of life organization, as well as of complexity.

### 6.3. From Diversity to Complexity: Interplay between Germline Cell DNA and Somatic Cell Phenotype

As depicted above, emergence through the course of evolution of molecular innovations (e.g., intracellular membranes, introns, and histones) impacting on cell and genome organization could have allowed an increase in the diversity of metabolic processes driven by one single genome (Figure 5E). As a consequence, genomes would have been exposed to an increasing diversity of potentially genotoxic biochemical compounds. In this setting, it has been proposed that meiosis protected DNA from cell metabolic activities. This hypothesis, known as the “dirty work hypothesis” [236,237], corresponds to the fact that meiosis relies on homologous recombination, a mechanism that removes damaged (e.g., oxidized) nucleotides (see Section A.3). In this model, meiosis is a process that removes damaged nucleotides while allowing the formation of haploid germ cells. Thanks to the formation of germ cells, the DNA transmitted across generations is no longer directly exposed to metabolic activities (i.e., the “dirty work”) yet still “benefit” from these activities [236,237]. By considering that the phenotype that depends on the germ cell DNA corresponds to the production of somatic cells, the relationship between a genome and its phenotype depicted at the molecular level (see Section 4) still operates in multicellular organisms. Indeed, germ cell DNA (the genome) gives rise to somatic cells (the phenotype), those activities allow the integrity of the germline DNA molecule to be maintained. The germline DNA transmitted to the next generation after fertilization allows the same phenotype to be generated under a stable environment (Figure 5F); note here that the “same phenotype” should not be understood in a literal sense (see Discussion). Although speculative, the molecular mechanisms establishing a continuum between physiological and genetic adaptation (depicted in Section 5) could still operate in multicellular organisms, since specific germ cell DNA locations or their associated histones could be biochemically modified when somatic cells challenged by environmental fluctuations produce molecules (e.g., RNAs) targeting specific genomic locations of germ cell DNA (see Section 5, Figure 4F,G).

The interplay between somatic cell phenotype and germ cell DNA stability can be illustrated by adaptation of multicellular organisms to cold, a process that can also illustrate several interesting relationships between the different scales of life organization. Indeed, cold can lead to genetic instability by decreasing the kinetics of (bio) chemical reactions, and this cold-induced genomic instability could be relaxed by increasing the activity of genes encoding enzymes involved in cellular respiration, and therefore increasing heat production [238]. However, increasing cellular respiration increases the production of ROS that induces genetic instability, and therefore can lead to increased activity of genes involved in the detoxification of ROS, such as those coding for uncoupling proteins (UCPs) [239,240] (Figure 5G, 1 and 2). Indeed, uncoupling proteins such as UCP1 mediate proton leaks across the inner mitochondrial membrane, which (i) mitigate ROS production and (ii) simultaneously lead to cellular heat production [239,240]. The UCP-dependent heat production has been proposed to contribute to the emergence of heat-producing muscle cells, and, next, the emergence of mammalian brown adipose cells that (i) express UCP1, (ii) derive from skeletal muscle progenitor cells, and iii) play an important role in heat production in mammals [241,242] (Figure 5G, 2 and 3). Interestingly, it has been proposed that the loss of genes such as UCP1 in bird ancestors (due to yet unknown mechanisms) did not allow the emergence of brown adipose cells but instead led to hyperplasia of heat-producing muscle cells. Thermoregulation depending on muscle hyperplasia (vs. brown adipose cells) has been proposed to generate constraints during development, with consequences on the body plan organization of birds (vs. mammals) [243,244] (Figure 5G, 4). 

To summarize, environment-dependent physicochemical parameters (e.g., cold temperatures) could create constraints on somatic and germ cells, trigger genetic instability, and lead to the emergence of new polymers (e.g., UCPs) and new cell types (e.g., muscle cells). Supporting this model, it has recently been shown that exposure of parents to cold induces epigenetic modifications in sperm with consequences on adipose tissue activity in the offspring [245]. Adaptation to cold also illustrates the interdependence between physicochemical properties at the molecular level (e.g., proton leakage) and physicochemical properties at the upper level of life organization (e.g., heat production by cells) that can relax the initiating constraints (Figure 5G). Adaptation to cold also illustrates that adaptation to environmental physicochemical parameters can generate side effects as well as diversity. For example, the emergence of specialized cells producing large amounts of energy that allow control over organism temperature would also allow the organism mobility to be improved, and different innovations (e.g., brown adipose tissue vs. muscle hyperplasia) have different consequences in terms of body plan. Thus, although first arising as a side-effect, emerging properties (e.g., mobility) would also be under the control of natural selection.

## 7. Conclusions

To summarize the proposed model, a genome in a stable environment generates a phenotype that maintains the stability of its originating genome, and both (genome and phenotype) are reproduced identically (Figure 6A, left panel). Obviously, the word “identical” should not be taken in the strictest sense, as sequence variations within genomes may not have major influence on the phenotype, and variations in phenotypes allow the same range of physicochemical constraints to be relaxed. In other words, a range of genotypes correspond to a range of phenotypes, which can cover a range of environment-dependent physicochemical constraints (Figure 6B). However, outside a physiological range of physicochemical parameters, a genome generates a phenotype that no longer maintains the stability of its originating genome and instead triggers mutations, whose rate, nature, and location dependent on the initial constrains and the challenged phenotype. This process occurs until new genetic variants generate a phenotype that maintains the stability of its originating genome (Figure 6A, right panel). It is important to stress that it is not a question of overall genome stability but of stability of genomic regions (for example, regions hosting certain genes) that are challenged by environmental fluctuations. 

In conclusion, while the notions of chance and natural selection are useful to highlight the fact that life has not been “created” by or for something, they cannot be considered as evolutionary driving forces. Instead, evolutionary driving forces correspond to environment-dependent physicochemical constraints that challenge the phenotype and the underlying genome, and thereby direct their evolution. Such evolutionary driving forces cannot explain all the diversity of life for several reasons. The first is that mutation processes based on physicochemical processes are probabilistic. This means that diversity can still emerge from chance occurrences (Figure 6C). Secondly, mutations that relax the initial constraints allow the emergence of new characteristics or properties as side effects and contribute to the diversity of living organisms [246] (Figure 6C, 2 and 3). Finally, the emergent properties resulting from molecular innovations under the constraint of physicochemical parameters can also generate new properties that can confer certain advantages and disadvantages at the organismal level. This means that the probability of dissemination within a population of some mutations at the origin of molecular innovations can potentially be modulated by various phenomena, including natural selection (Figure 6C). Therefore, the notions of random mutations and natural selection are not evolutionary driving forces but contribute to life form diversity and act as a filter, respectively.

## Figures and Tables

**Figure 1 life-10-00007-f001:**
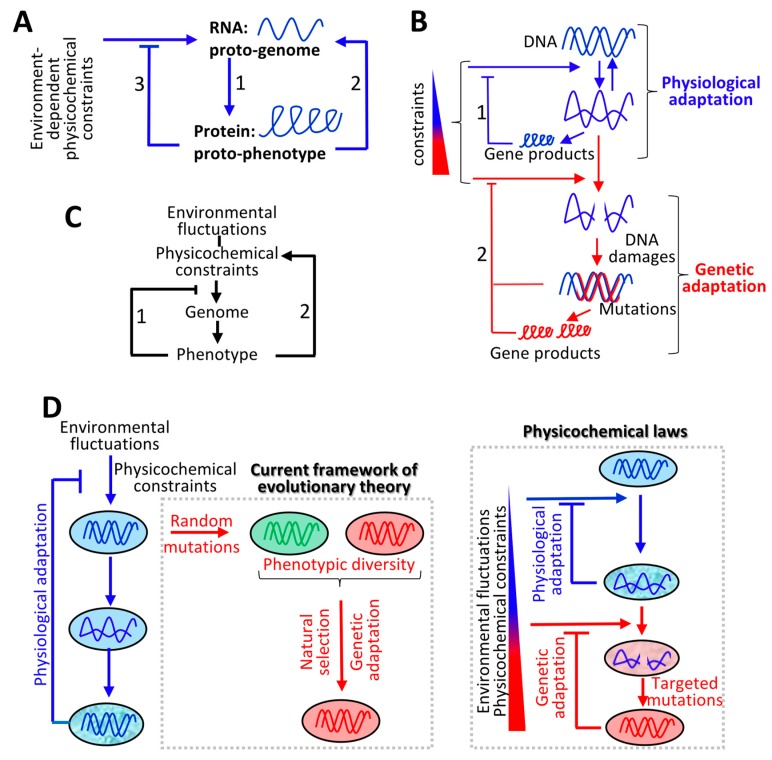
(**A**) The origin of life probably relies on simpler forms of organization than those observed in modern living organisms. The commonly accepted hypothesis postulates that the origin of life corresponds to the emergence of polymers, such as RNAs and proteins. These two polymers are interdependent as RNAs serve as templates for protein synthesis (1) and proteins are necessary for RNA synthesis (2). This interdependence, which can be represented in the form of feedforward and feedback loops between the proto-genome (RNAs) and the proto-phenotype (proteins) can only be maintained if proteins relax environment-dependent physicochemical constraints triggering for example RNA degradation (3). This interdependence is the foundation of life and evolution. (**B**) Variations in the extracellular environment induce constraints on cellular polymers sensitive to these variations, triggering the cellular response that results in the biogenesis of gene products, whose activity relaxes the initial constraints (1). However, if these variations exceed a certain amplitude or persist over time, they challenge the integrity of the targeted polymers, which ultimately lead to mutations. This process stops only when new sequences (directly or indirectly) relax the initial constraints (2). (**C**) A DNA molecule is subjected to environmental fluctuations of physicochemical parameters, which triggers the biogenesis of polymers (gene products) whose activities correspond to the phenotype. Cellular activities (the phenotype) allow a return to equilibrium by relaxing the initial constraints (1). Nevertheless, these activities also generate constraints directly or indirectly on their originating genome, meaning that a genome is adapted to the constraints generated by its own activities (2). (**D**) According to the current framework of evolutionary theory (left panel), there is no direct relationship between physiological and genetic adaptation because physiological adaptation is based on physicochemical principles of homeostasis as a function of environmental fluctuations, while genetic adaptation would be fueled by random mutations generating a diversity of phenotypes on which natural selection acts. In contrast, in the model proposed in this article (right panel), genetic adaptation is the consequence of physiological adaptation. Indeed, physiological adaptation can take place as long as fluctuations in environment-dependent physicochemical parameters do not exceed a certain threshold. Above this physiological threshold, the integrity of nucleic and amino acid polymers, in particular DNA, is challenged which leads to targeted mutations. This mutational process stops when the mutations generate a phenotype that maintains the integrity of the DNA with regard to environmental constraints (genetic adaptation).

**Figure 2 life-10-00007-f002:**
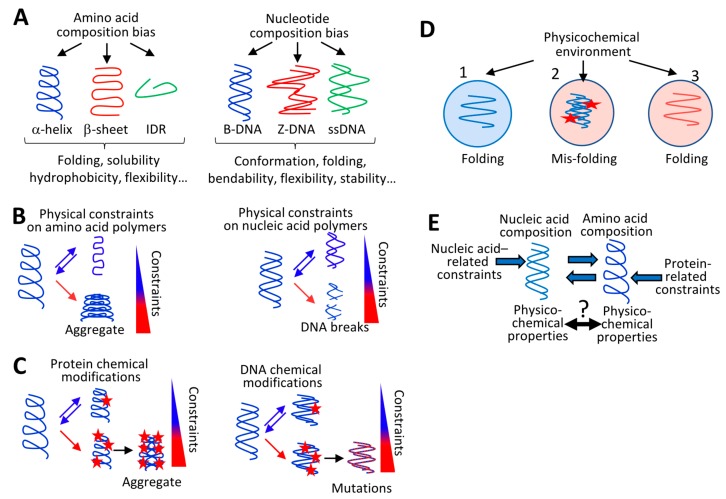
(**A**) The amino acid or nucleotide composition of proteins (left) or DNA (right), respectively, determines the physicochemical properties of these polymers with consequences on their physical and chemical properties. IDR, intrinsically disordered region and ssDNA, single-stranded DNA. (**B**) The composition of a polymer determines its physicochemical properties, and therefore its folding and physical resistance to specific constraints. Depending on the composition of a given polymer, some physicochemical constraints induce reversible structural changes (blue arrows), and others induce irreversible damages (e.g., aggregation and breaks) (red arrows). (**C**) Chemical modifications of amino acids (left) or nucleotides (right) change the physicochemical properties of polymers. These chemical modifications are reversible (blue arrows) or induce irreversible damages (red arrows). (**D**) Any given polymer (blue) is stable in a physicochemical environment and unstable in another one (1 vs. 2). A different polymer (red) reacts differently under the same constraints (3 vs. 2). (**E**) Composition determines the physicochemical properties of nucleic or amino acid polymers. Nucleotide or amino acid composition is constrained by physicochemical parameters, which suggests that their composition must correspond to the same fundamental physicochemical parameters as the sequence of nucleic acid polymers determines the composition of proteins.

**Figure 3 life-10-00007-f003:**
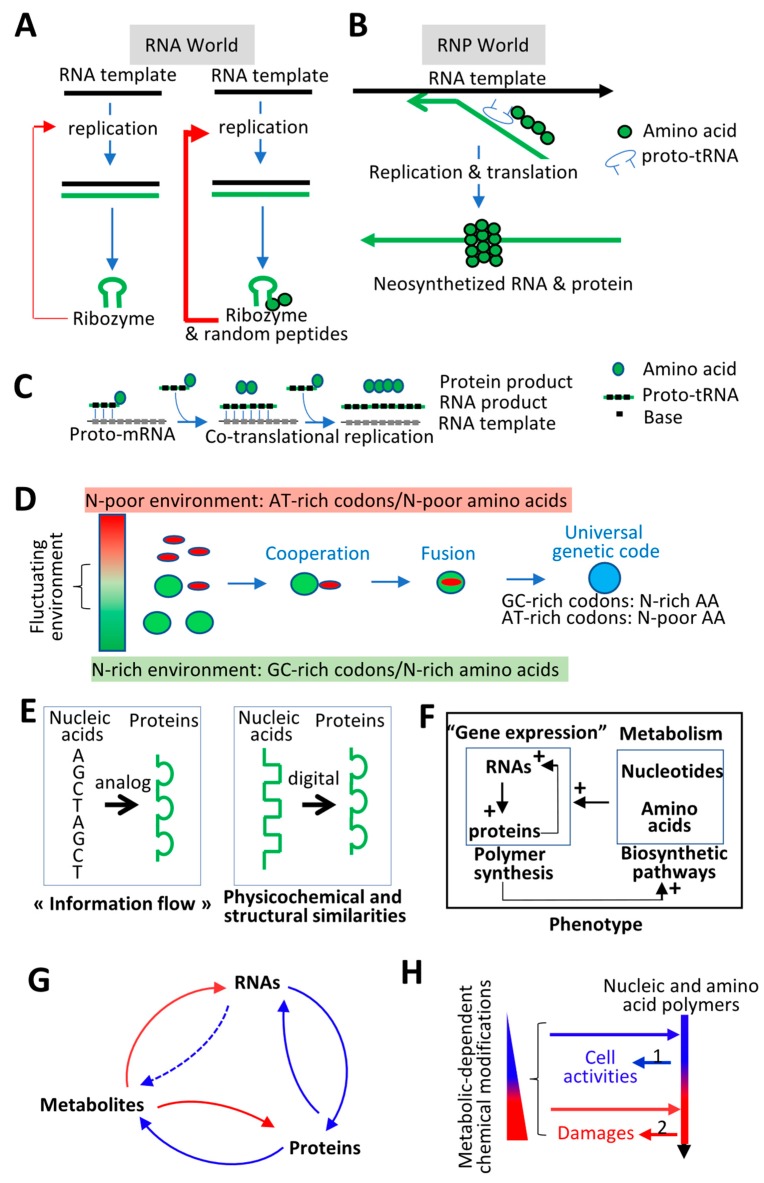
(**A**) In an RNA world, an RNA molecule is replicated thanks to the product of replication (e.g., ribozyme, left panel). This process is enhanced by amino acids or small peptides (right panel). (**B**) In an RNP world, replication (RNA production) and translation (protein production) could have been performed simultaneously as in modern prokaryotes, thereby avoiding the nascent RNA to interact back to the template and increasing the probability that the neo-synthetized protein interacts with the RNA replication product. (**C**) Cotranslational replication in an RNP world could have be performed owing to amino acids attached to proto-tRNAs that enhanced the polymerization of proto-tRNAs and that were simultaneously incorporated into the nascent protein. (**D**) Two different proto-cells (red and green circles) growing in different chemical environments (e.g., N-poor vs. N-rich environment) could have developed different proto-genetic codes. Their cooperation in fluctuating environments could have led to horizontal transfer, leading to the emergence of the universal genetic code. (**E**) Nucleic acid polymers are often represented as linear strings of letters that are translated into proteins with physicochemical properties unrelated to those of nucleic acid polymers. If the genetic code has been constrained over evolutionary time to match nucleic acid polymers and their cognate amino acid polymers to the same fundamental physicochemical constraints (e.g., temperature, element availability), nucleic and amino acid polymers share more physicochemical properties than previously anticipated. (**F**) In an RNP world, cooperation between interdependent polymers (i.e., RNAs and proteins) relies on their activities toward the biogenesis of nucleotides and amino acids necessary for their synthesis. The phenotype of a proto cell in an RNP world corresponds to the polymerization of nucleotides and amino acids. The polymerization products produce nucleotides and amino acids. (**G**) RNAs give rise to proteins, which allow the biogenesis of RNAs and metabolites through transformation of molecules captured from the environment (blue lines). RNAs probably played a role, at least early in evolution, in metabolite biogenesis (blue broken lines). Metabolites are required, in turn, to give rise to RNAs and proteins (red lines). (**H**) Metabolic-dependent chemical modifications of nucleic and amino acid polymers contribute to the cell activities and to the cellular physiological adaptation in response to environment-dependent constraints (1). However, metabolic-dependent chemical modifications can also induce irreversible damages of nucleic and amino acid polymers (2).

**Figure 4 life-10-00007-f004:**
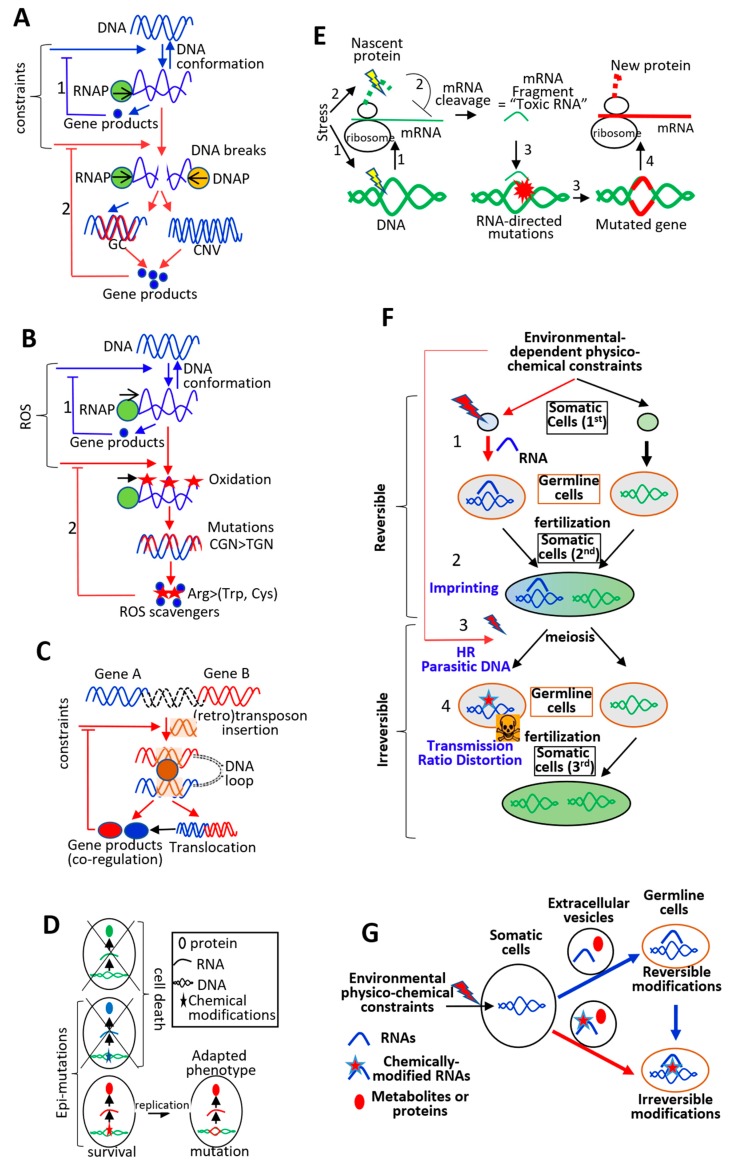
(**A**) Variations in the extracellular environment increase the expression of target genes which is a process associated with physical constraints on DNA generated by RNA polymerases (RNAP). These physical constraints are transient if the gene activity relaxes the initial constraints (e.g., through the biogenesis of gene products) (1). If not, physical constraints persist, and conflicts between RNAP and DNA polymerases (DNAP) induce DNA damages. DNA damage is repaired by homologous recombination, which favors GC over AT nucleotides, but which can also induce gene copy number variation (CNV). Both of these processes, in turn, generate more gene products, and therefore relax the initial constraint (2). (**B**) Cellular stresses activate the expression of specific target genes while increasing the production of intracellular ROS. One strand of transcriptionally induced genes is exposed to ROS, promoting deamination of methyl cytosine, which gives rise to thymine. C > T mutations change Arg codons into Trp and Cys codons. Trp- and Cys-containing proteins can play a role in protecting cells from ROS, therefore, relaxing the initial constraints (2). (**C**) Transcriptional activation of genes induces neo-insertion of repeated sequences within transcription-dependent ssDNA. Neo-insertion of repeated sequences can facilitate co-regulation of two genes by bringing them closer to each other in space, and it also promotes recombination. Both processes coordinate the production of gene products and contribute to relaxing the initial constraints. (**D**) Cellular stress induces chemical modifications of target genes, which affects chromatin organization and transcription fidelity (“epi-mutations”). Chemical modifications that induce biogenesis of new RNAs and proteins could allow survival of the cells in which these modifications took place. Chemical modifications in surviving cells lead to mutations that increase the survival probability of daughter cells. (**E**) Cellular stress induces physical constraints simultaneously on target genes and on proteins produced from these genes (1 and 2). By disrupting translation, for example, by inducing nascent protein unfolding, a stress can induce translation stopping and cotranslational cleavage of mRNAs. RNA fragments generated during translation or transcription hybridize on the complementary DNA strand and locally generate DNA or chromatin modifications, thus, increasing the probability of mutations occurring in the targeted regions (3). This process stops only when the gene and its products obtain physicochemical properties that relax the initial constraints (4). (**F**) Somatic cells are constrained by environmental fluctuations, and their activities can have consequences for germ cells, for example, through the transfer of metabolites or small RNAs from somatic to germline cells (1). These compounds can change the activity of germ cells, affect the development of the body after fertilization, and cause mutations in germ cells (2 and 3). These mutations could lead to the emergence of somatic cells (3 and 4) whose activities maintain the integrity of the germ cell genome of the following generations. (**G**) Somatic cell genes that are constrained by environmental parameters produce extracellular vesicles containing RNAs that can either induce reversible epigenetic mutations or irreversible genetic mutations.

**Figure 5 life-10-00007-f005:**
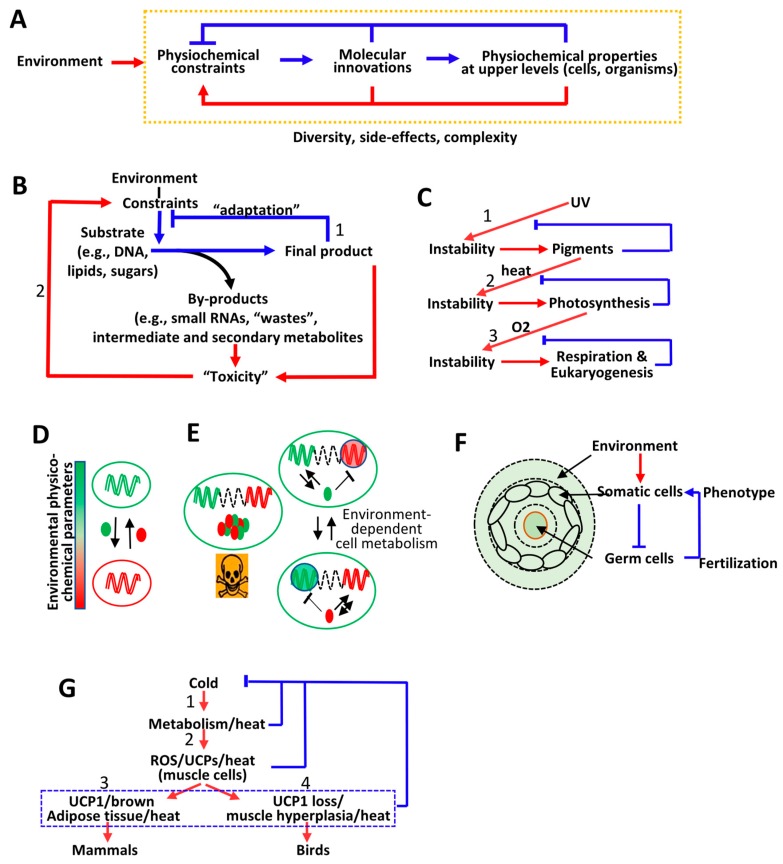
(**A**) The environment generates physicochemical constraints which, by destabilizing genomes, induce molecular innovations, allowing the emergence of new physicochemical properties at different scales of life organization. These molecular innovations and new properties are transmitted across generations if they release the initial constraints (blue lines) but can also generate new constraints (red lines), allowing the adaptation of the genome to its own activities and generating an evolutionary dynamic. (**B**) Any biochemical reaction leads to the synthesis of a final product (1) and “waste”, by-products, or secondary metabolites. These compounds are not necessarily essential to the cell’s survival, but they are the result of vital cellular activities. Therefore, these compounds are not “random products” even though they can have cellular toxic effects by interacting with cellular polymers (2), which can lead to mutations. (**C**) UV radiations induce genomic instability favoring genomes that produce pigments that, in turn, protect them from the initial constraint (1). Likewise, UV radiation-absorbing pigments induce genomic instability favoring genomes that produce photosynthetic reaction centers that, in turn, protect them from the initial constraint (2). Likewise, O_2_ production by photosynthesis induces genomic instability favoring genomes that produce oxidases and cellular components that, in turn, protect them from the initial constraint (3). (**D**) Two unicellular organisms (green and red), which are adapted to different environment-dependent constraints, cooperate by exchanging various components in a fluctuating environment. (**E**) Not all enzymatic reactions generated from a genome can take place simultaneously (left panel). The selective compaction of different regions of a genome according to the cellular metabolic state (and therefore its environment) through chemical modifications of histones protecting some genome parts and repressing their potentially toxic expression while allowing the expression of genes whose products contribute to maintain the cellular homeostasis. (**F**) Somatic cells “buffer” environment-dependent constraints, maintaining the stability of the genome from the germ cells. When “protected” by somatic cells (i.e., when the phenotype of the organism is adapted to its environment), germ cells give rise to gametes that generate after fertilization the same somatic cells, i.e., phenotype. (**G**) Cold induces physiological adaptation by activating cellular respiration that increases cellular heat production (1). However, an increase in cellular respiration increases ROS production, which can lead to activation of UCP proteins. This simultaneously balances the ROS genesis and increases heat production in muscle cells (2) or in brown adipose tissue (3). The loss of UCP1 in bird ancestor may have led to muscle hyperplasia for heat production with consequences on bird body plan (4).

**Figure 6 life-10-00007-f006:**
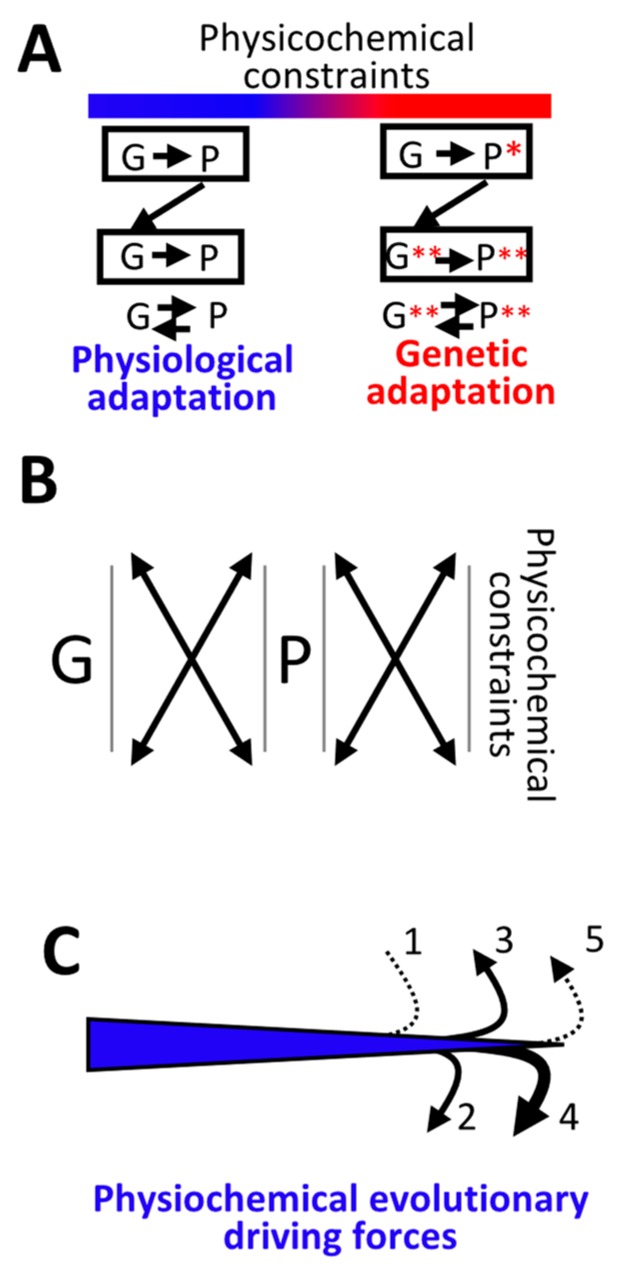
(**A**) A genome (G) generates a phenotype (P) that maintains the stability of its originating genome in a stable environment and both (genome and phenotype) are reproduced identically (left panel). In an unstable environment (corresponding to variations in physicochemical parameters above a physiological range), the genome generates a phenotype (P*) that no longer maintains the stability of its originating genome and instead triggers mutations whose rate, nature, and location dependent on the initial constrains and the phenotype (right panel). This process occurs until new genetic variants (G**) generate a phenotype (P**) that maintains the stability of its originating genome. (**B**) In a given environment, a genome (G) gives rise to a range of phenotypes (P), and similar phenotypes can correspond to a range of genomic sequences. Each phenotype can be adapted to a range of environmental physicochemical constrains. (**C**) Evolutionary driving forces rely on physicochemical processes whose probabilistic nature generates genetic and phenotypic diversity, as symbolized by arrows from 1 to 5. If a path (exemplified by Path 1) does not stabilize its originating genome, the genome and the corresponding phenotype will not be reproduced. While some paths might be neutral in terms of natural selection (Paths 2 and 3), some paths (Paths 4 and 5) could lead to the emergence of phenotypes that can be under natural selection.

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
