# Peer review of "Physicochemical Foundations of Life that Direct Evolution: Chance and Natural Selection are not Evolutionary Driving Forces"

_life, 2020, doi:10.3390/life10020007_

Round 1

Reviewer 2 Report

Many of the assertions made in this paper are original, have merit, and are of importance. But this author extrapolates his valid insights on how the interplay of protein and nucleic acid evolution constrain one another physicochemically to explain every aspect of organismal evolution. With the more modest objective of showing how current knowledge of molecular evolution undermines the Modern Synthesis and dominance of random-search natural selection this paper could be very influential. But as a purported "theory of everything" it comes up against too many counterexamples and would probably be dismissed by most readers.

Just to take the example of multicellularity, something I am familiar with: a multiplicity of forms arise in both animals and plants because of the physical nature of the multicellular materials. These properties are very different in the two groups, and the forms are predictably different. The organismal phenotypes do not mainly arise, or serve to, protect the integrity of the respective genomes as claimed. Morphological phenotypes can vary with little or no genetic change (plasticity) and genomes can change with no morphological change (developmental system drift).

Many other cases, particularly in Part 4, involving evolution of cellular features like histones and their chemical modifications, photosynthesis, respiration, gametogenesis, ecosystems, and so forth, really seem to be stretching the author's main idea. There may be grains of truth in some of these scenarios, but some are farfetched. Multicellularity has emerged in more than a dozen lineages, based on different cell surface molecules. Sometimes it is based on cells not separating after division, sometimes it is due to aggregation. In some cases, a cell surface molecule that is not adhesive in one environment becomes sticky in a different one. This is all physicochemistry, but is disconnected, or very distant, in most cases from the requirement that a new phenotype primarily serves to stabilize the originating genome. Often phenotypic innovations are just side-effects of other processes in slightly changed environments, and the new phenotypes are "neutral" in John Tyler Bonner's sense.

The author has presented an interesting perspective on how the Darwinian theory conflicts with coevolution of DNA and protein structure. My recommendation would be to reserve the material in Part 4 for later publications, insofar as the case can be made for any of the examples, and revise this contribution to just focus on molecular evolution (Parts 1-3).  

Round 2

Reviewer 2 Report

On line 1026 the author states "Indeed, uncoupling proteins, such as UCP1, are anion carriers that mediate proton leaks across the inner mitochondrial membrane..." I found this confusing, since protons are not anions. A literature search showed that the author is technically correct, since UCP1 is structurally part of a family whose typical role is to transport anions, and even UCP1 appears to transport some anions. But since the role of UCP1 in generating heat is not anion transport, it might be desirable to leave out this family designation.

Author Response

The definition of UCP proteins as anion transporters has been removed as requested.